# The Role of Autophagy in Type 2 Diabetic Kidney Disease Management

**DOI:** 10.3390/cells12232691

**Published:** 2023-11-23

**Authors:** Che-Hao Tseng, Kavya M. Shah, I-Jen Chiu, Li-Li Hsiao

**Affiliations:** 1Renal Division, Department of Medicine, Brigham and Women’s Hospital, Harvard Medical School, Boston, MA 02115, USA; b101106003@tmu.edu.tw (C.-H.T.); kavyashah@college.harvard.edu (K.M.S.); 2Division of Nephrology, Department of Internal Medicine, Shuang Ho Hospital, Taipei Medical University, New Taipei City 23561, Taiwan; 3Department of Internal Medicine, School of Medicine, College of Medicine, Taipei Medical University, Taipei 11031, Taiwan; 4TMU-Research Center of Urology and Kidney (TMU-RCUK), Taipei Medical University, Taipei 11031, Taiwan

**Keywords:** autophagy, diabetic kidney disease, diabetic nephropathy

## Abstract

Diabetic kidney disease (DKD), or diabetic nephropathy (DN), is one of the most prevalent complications of type 2 diabetes mellitus (T2DM) and causes severe burden on the general welfare of T2DM patients around the world. While several new agents have shown promise in treating this condition and potentially halting the progression of the disease, more work is needed to understand the complex regulatory network involved in the disorder. Recent studies have provided new insights into the connection between autophagy, a physiological metabolic process known to maintain cellular homeostasis, and the pathophysiological pathways of DKD. Typically, autophagic activity plays a role in DKD progression mainly by promoting an inflammatory response to tissue damage, while both overactivated and downregulated autophagy worsen disease outcomes in different stages of DKD. This correlation demonstrates the potential of autophagy as a novel therapeutic target for the disease, and also highlights new possibilities for utilizing already available DN-related medications. In this review, we summarize findings on the relationship between autophagy and DKD, and the impact of these results on clinical management strategies.

## 1. Introduction

Diabetic kidney disease (DKD), or diabetic nephropathy (DN), is a common but severe complication in diabetic patients. The disorder mainly consists of persistent proteinuria with progressive worsening of renal function, and can ultimately cause irreversible kidney damage [1]. As the leading cause of end-stage kidney disease (ESKD), DKD is associated with increased morbidity and mortality in diabetes patients, and thus poses a considerable threat to healthcare systems across the globe. It has been estimated that of the global population of type 2 diabetes mellitus (T2DM) patients, over 35–40 percent eventually develop DKD [2]. Several risk factors for DKD have been identified, most of which are related to the underlying metabolic syndrome. Genetic components also play a role in the development of the disease, and a persisted hyperglycemic state remains an important contributing factor to the progression of DKD [3]. The pathophysiological mechanism of DKD arises from the interaction between various metabolic pathways, including reactive oxygen species (ROS) and advanced glycation end-products (AGEs) [2,4,5]. This mechanism triggers abnormal downstream cellular responses, which in turn lead to gross vascular injury and renal cell damage. 

Considering the high prevalence of DKD and the large population of T2DM patients at risk, updates on the understanding and management of the disorder are essential. Clinically, blood glucose/pressure control and renin-angiotensin-aldosterone system (RAAS) medications are supported by strong evidence for halting the development and progression of DKD. However, much is unknown about the underlying mechanism and stage-specific strategy for DKD. Over the years, researchers have identified numerous pathways that are related to the pathophysiology of DKD, with oxidative stress and metabolic imbalance caused by a persisted hyperglycemic state taking center stage. These pathways include the mitogen-activated kinases (MAPK) pathway, which leads to podocyte apoptosis and extracellular matrix deposition, and pathways that are heavily involved in inflammation, such as the Janus kinase-signal transducers and activators of transcription (JAK/STAT) and nuclear factor kappa-B (NF-κB) pathways [3]. Together, these pathways contribute to the complex regulatory network of DKD.

Autophagy is known to be an important mechanism for maintaining intracellular homeostasis. The process plays a wide variety of physiological roles, and dysregulation of autophagy can cause irreparable tissue damage [6]. There are three subtypes of autophagy: microautophagy, macroautophagy, and chaperone-mediated autophagy. Macroautophagy is the most prevalent form of the three, and thus will be referred to as autophagy hereafter unless otherwise specified. The process of autophagy can be divided into several stages: first, the initiation of autophagy due to stress stimulation; next, the formation of phagophores and encapsulation of intracellular components into autophagosomes; and last, the fusion of autophagosomes with lysosomes, which form autophagolysosomes that ultimately go through the degradation process [7]. Multiple genes and signaling pathways can interfere with this process, and are closely related to the microenvironment the cells are in.

In light of its role in various diseases, increasing evidence indicates that the regulation of autophagy plays a complex role in the progression of DKD, and conversely that the modulation of autophagy in DKD may have potential as a therapeutic target in the future. In this review, we aim to explore the most recent understanding of the role of autophagy in DKD progression, while establishing the relationship between autophagy and current clinical management strategies and medications.

## 2. The Role of Autophagy in Kidney Diseases

The physiological role of autophagy is moderated by a complex network of signaling events, along with more than 30 autophagy-related (Atg) proteins that regulate the formation of phagophores and autophagosomes [8]. The formation of phagophore membranes in mammalian cells appears to be in dynamic equilibrium with other cytosolic membrane structures that primarily originate from the endoplasmic reticulum. The Atg protein family regulates various processes related to autophagic initiation, such as lipid recruitment and increased sensitivity to growth factor and nutrient availability [9]. Autophagy regulation is critical in various diseases as it is related to inflammatory and fibrotic responses during cellular stress [10]. In the case of kidney diseases, with the aid of computational gene transcriptional pathway analysis, responsive autophagy was found to be activated in multiple types of glomerular injury, including membranous nephropathy (MN), focal segmental glomerulosclerosis (FSGS), IgA nephropathy (IgAN), and especially in DN [11]. However, autophagy seems to play dual, somewhat opposing, roles in kidney disease. In acute kidney injury models, it has been suggested that autophagy plays a physiological role in inhibiting excessive inflammatory response, therefore alleviating early kidney injuries, explaining the aforementioned reactive activation of autophagy observed [12]. However, in studies concerning the later stages of kidney fibrosis, or in a proposed acute kidney injury (AKI) to chronic kidney disease (CKD) transition model, dysregulated autophagy was associated with uncontrolled fibrosis and further kidney function decline [13]. The role of autophagy regulation in kidney disease is an intriguing topic that requires more investigation. 

Regarding the role of autophagy in DKD, most studies suggest that a decline in autophagic activity can be observed at the peak of DKD progression, so autophagy has been seen as a primary target for developing novel means of intervention [14]. Functioning autophagic processes are essential to preventing the development of DKD, as autophagy allows kidney cells to remove damaged organelles and proteins accumulated from prolonged cellular stress. Autophagy is known to be connected to the pathophysiological components of diabetes and subsequent complications. Besides mediating the oxidative stress and ROS generated by diabetes, autophagy also inhibits the accumulation of AGEs, a key element involved in the development of DKD [15]. Additionally, under a prolonged hyperglycemic state, the deactivation of nutrient-sensing pathways such as the AMP-activated protein kinase (AMPK) and the downstream mechanistic target of rapamycin (mTOR) pathways [16,17] can lead to decreased autophagic activity. Fewer quantities of phosphorylated(p)-AMPK, the activated form of AMPK, have been found in a high-glucose state, which leads to increased mTOR activity and inhibits the typical induction of autophagy and formation of autophagosomes [18]. In this manner, impaired autophagic regulation contributes to the development of DKD. 

In the following sections, we will summarize the most recent updates on established autophagic regulatory pathways, as well as how the regulation of autophagy can affect the progression of DKD (Table 1).

## 3. The Mechanistic Basis for Autophagy in DKD

### 3.1. An Increased Understanding of Nutrient-Sensing and Downstream Pathways

Due to a constant hyperglycemic state, the nutrient-sensing AMPK and mTOR signaling pathways have established their role in influencing DM-induced pathological changes [26,27,28]. The downregulation of AMPK signaling in DN leads to decreased autophagic activity and worse outcomes [29], while the mTOR signaling pathway, which is overactivated in DN, blocks the initiation of autophagy [30,31]. Recently, the JAK/STAT signaling pathway has been shown to be closely related to energy expenditure while also being regulated by AMPK signaling [32,33]. Previous studies have demonstrated the potential of hindering DN development via regulating JAK/STAT signaling; the pathway’s effect on autophagy regulation has also been shown in chronic myeloid leukemia cells [34,35]. However, reports regarding the impact of the JAK/STAT pathway on autophagy in kidney cells is lacking. A recent update by Chen et. al. found that in a DKD mice model the JAK/STAT signaling pathway was activated in podocytes under hyperglycemic conditions, and MAP1LC3 (LC3) expression, which serves as a proxy for autophagic activity, was downregulated, leading to subsequent cellular apoptosis and DKD progression [19]. By inhibiting the JAK/STAT signaling pathway in cell models via the JAK inhibitor ruxolitinib, impaired autophagic flux was restored. Another pathway closely related to nutrient-sensing, the p53 signaling axis, was recently introduced as a regulator of autophagic signaling [36]. Activated p53 signaling under DKD models was found to impair autophagy response in renal tubular cells via the unc-51-like autophagy-activating kinase 1 (ULK1) pathway [20], and in podocytes via the nutrient-sensing Sirtuin-1 (SIRT1) pathway, leading to subsequent DKD progression [21]. Together, these results indicate that impaired autophagic activity from dysregulated nutrient-sensing pathways can promote the development of DKD.

Researchers are still looking to further investigate the cellular response to hyperglycemic states, specifically with regards to the involvement of common energy-containing nucleotides such as adenosine triphosphate (ATP), adenosine diphosphate (ADP), uridine triphosphate (UTP), and uridine diphosphate (UDP) [22]. Under hyperglycemic conditions, these nucleotides can be released extracellularly, and can subsequently trigger cellular purinergic P2 (P2Y) receptors. Among the eight known subtypes of P2Y receptors, P2Y2R was found to be prominently expressed in renal glomerular, mesangial, podocyte, and tubular cells. These updates expand the known scope of nutrient-sensing pathways, demonstrating their involvement in the autophagy-DKD axis. 

### 3.2. Autophagic Activity Affects Macrophage Phenotypic Change

The current understanding of upstream pathways of DKD helps in identifying potential therapeutic targets, as well as predicting the effect of interrupting the involved signaling axes. Additionally, the downstream effects of autophagy and how the process complicates the development of DKD progression is of equal significance, as this information can assist researchers in identifying the various effects of autophagic regulation. 

Within the pathophysiology of DKD, the dysregulation of autophagic activity itself contributes to the impaired repair of renal cells under stress. Another important physiological repair mechanism arises from the effect of macrophages [37]. The migration and adhesion of macrophages to diabetic kidney tissue is known to play a major role in DKD progression, as it leads to the release of cytokines and chemokines, which subsequently induce inflammatory damage to the kidney. Interestingly, studies have connected macrophage functionality with autophagic activity. It was found that a hyperglycemic state not only affects autophagic activity in kidney cells but also reduces the number of autophagosomes in RAW264.7 cells, a mouse macrophage cell line [23]. The inhibition of autophagy in macrophages was accompanied with an increased expression of P62, which promoted the adhesion and migration capacity of macrophages. This same pattern of autophagic and P62 upregulation can also be seen in the renal tissues of a DN rat model. This is also, in part, related to the fact that macrophage autophagy can inhibit M1 pro-inflammatory macrophage polarization, which, in turn, can alleviate the progression of tissue damage [24]. M2 macrophages were also found to be able to secrete autophagy-activating exosomal miR-25-3p, and ameliorate high glucose-induced podocyte injury, adding to the complex relationship between macrophages and autophagy [38]. Yuan et al. reported that the interaction between autophagy and macrophage phenotypic change may be the underlying mechanism behind the beneficial effects of mesenchymal stem cell (MSC) treatment [39]. By treating DN mice via MSC vein infection, their data showed a restoration of autophagic activity and an inhibited inflammatory response in macrophages, which alleviated renal injuries. The authors also identified activation of transcription factor EB (TFEB) as the mechanism behind the effects of the MSC treatment. TFEB nuclear localization was significantly increased after MSC treatment in RAW 264.7 cells, which elicited the polarization of M2 macrophages. A better understanding of the interactions between autophagy, DKD, and macrophage function may provide us with a fresh outlook on the effects of autophagy outside of kidney tissue, and the other intercellular events that are a part of DKD progression.

### 3.3. New Prospects in Gene and Epigenetic Regulation

The final aspect concerning the regulatory pathways of autophagy in DKD that we wish to highlight in this section is the role of epigenetics. With precision medicine being the future trend in healthcare, the recognition of the specific genetic components of disease has become increasingly crucial. Epigenetic mechanisms such as histone modifications and microRNA regulation are believed to assist in physiological adjustments to environmental conditions. Components of epigenetic modulation, including noncoding RNAs (ncRNAs), histone deacetylases (HDACs), and DNA methylation, were all found to play a unique role in DKD progression [40,41]. Studies have shown that persistent inflammation and cytokine exposure under diabetic conditions leads to epigenetic modifications, and ultimately induces lasting open chromatin structures at pathological gene sites, a possible mechanism for “metabolic memory” [42]. Numerous miRNAs are involved in autophagic suppression under diabetic conditions, with some achieving their effect by taking part in some of the aforementioned signaling events [43,44]. Reversing the effects of these miRNAs, then, is expected to reduce glomerular mesangial hypertrophy in early stages of DN in animal models, which may be a viable therapeutic strategy in the future.

In addition to the effect of miRNAs, HDACs have also been identified as an important family of enzymes that take part in numerous physiological processes, including diabetes and insulin resistance [45]. In the case of DKD, it was established that HDACs can promote fibrosis and chemokine production in kidney cells. Further research found that different isoforms of HDACs may also exert individual effects on autophagy in DKD [46,47]. HDACs 4 and 5 were among the earliest enzymes within the family that were found to play an inhibitory effect on autophagy; in fact, suppressed autophagic activity and increased inflammation were both observed. Adding to this connection, Liang et al. showed that HDAC6, an HDAC that is predominantly located in the cytoplasm and is related to the stability of microtubules, was activated in T2DM patients, db/db mice models, and advanced glycation end products (AGE)-treated podocytes [25]. Their further results indicated that the podocytes suffered from subsequently enhanced motility and suppressed autophagy, leading to DN progression, which could be rescued by tubacin, an HDAC6-specific inhibitor. A number of medications such as valproic acid—a widely used anti-epileptic drug—have been recognized as possessing HDAC-inhibiting effects [48]. Although specific medications that target epigenetic mechanisms may not currently be readily available, the advancement of pharmaceutical technology, tailored gene therapy, and epigenetic interventions may provide an exciting avenue for treating DKD in the future.

## 4. The Role of Autophagy within the DKD Clinical Regimen

Developing optimal management strategies against DKD is vital for DM patients. Current evidence-based clinical treatment guidelines suggest modifying existing treatment approaches using oral hypoglycemic agents (OHAs), such as metformin and sodium–glucose cotransporter 2 inhibitors (SGLT2i), based on albuminuria or low eGFR status [49,50]. For DM patients with albuminuria or/and hypertension, the prescription of renin-angiotensin-aldosterone system (RAAS) blockers is also strongly recommended. 

The role of autophagy in DKD progression has received increasing attention, and various studies have focused on the correlation between autophagy and DN-related medications. In the following section, we will summarize current findings and how they may affect the utilization of these agents (Figure 1). 

### 4.1. The Importance of OHAs in DM Management

In patients suffering from DKD, blood sugar control is highly important. Effective blood sugar control may not only stop disease progression, but may also prevent the development of other comorbidities [51]. With increasing patients being diagnosed with DM, OHAs are now the centerpiece of DM management. Common agents include metformin, sodium–glucose cotransporter 2 inhibitors (SGLT2i), dipeptidyl peptidase-4 inhibitors (DPP4i), and glucagon-like peptide-1 receptor analogs (GLP-1 RA), with each having different mechanisms of action. Since these agents have become broadly available, researchers have been working to understand their impact on metabolic disorders, as well as the underlying mechanisms behind each drug. With DKD being the most prevalent secondary complication from DM, more insights into the influence of OHAs on DKD is essential. 

#### 4.1.1. Metformin

Metformin as a first-line OHA

Evidence has shown that a rigorous control of blood glucose to near-normoglycemia levels is beneficial to diabetic patients, as this can delay the onset and progression of albuminuria and reduced-eGFR [52,53]. As a first-line agent, metformin is a safe and effective medication that has been shown to help improve patients’ Hemoglobin A1c (HbA1c), as well as body weight and cardiovascular mortality [54,55,56]. The use of metformin in patients with kidney diseases is limited due to the drug requiring renal clearance. Current FDA guidance states that metformin is contraindicated in patients with an eGFR < 30 mL/min/1.73 m^2^, and that the effects of the drug should be monitored when an eGFR < 45 mL/min/1.73 m^2^ is noted. It is currently believed that activation of the AMPK signaling pathway is the key mechanism behind the drug’s benefits [57]. This mechanistic action was first seen in liver cells, where enhanced insulin sensitivity and fat metabolism were observed as a result of AMPK activation.
Metformin as a renoprotective agent that restores autophagic activity

Since the AMPK pathway plays an important role in autophagic activity, the regulatory effects of metformin were reasonably assumed to revolve around assistance in restoring impaired autophagy in DN. Results from diabetic mice models agreed with this hypothesis, showing that, indeed, metformin enhanced defective autophagic activity during kidney injury [58,59,60,61]. These studies suggested that metformin induces several autophagic pathways, including the Sirt1/FoxO1 axis and the mTOR pathway, both in vitro and in vivo. The activation of autophagic pathways halted structural changes in glomeruli and preserved renal function after injury. Wang et al. also observed that metformin inhibited the epithelial-to-mesenchymal transition (EMT) in vitro, and tubulointerstitial fibrosis (TIF) in vivo [60]. These findings further validated the benefits of using metformin as a first-line DM medication. Similar autophagy-inducing effects of metformin were also reported in other disease models [62,63,64], primarily leading to the amelioration of inflammation, oxidative stress, and respective pathological progression, underscoring the significance of the interaction between autophagy and metformin. 

Our current understanding suggests that the involvement of metformin in autophagic regulatory pathways can help attenuate kidney damage in DKD. Despite its limited role in later stages of kidney damage due to renal clearance, the renoprotective effects of metformin via autophagy regulation may further elevate its importance in DN prevention.

#### 4.1.2. SGLT2, DPP4 Inhibitors and GLP-1 Analog

The multifaceted benefits of SGLT2i

SGLT2i are another class of OHAs that have been commonly prescribed in recent years, and have been recognized for their multifactorial benefits, especially with regards to their cardiovascular and renoprotective effects [65,66,67]. The primary function of SGLT2i, reducing blood glucose, is achieved via the inhibition of SGLT2 receptors in proximal tubular cells, which leads to a subsequent increase in glucosuria. The beneficial effects of SGLT2i in DKD and CKD have been demonstrated via several large clinical trials, with significant outcome improvements and risk reduction of ESRD and death from renal causes [68,69,70]. The significance of SGLT2i as a first-line agent for T2DM patients with CKD has been highlighted in various clinical practice guidelines, including the Kidney Disease Improving Global Outcomes (KDIGO) guidelines.
SGLT2i promote autophagy via glucose-independent effects

Among the many clinical benefits of SGLT2i, glucose-independent effects such as the moderation of intrarenal hemodynamics were identified as one potential mechanism of action [71]. Recent evidence suggests that restoration of autophagic activity by SGLT2i is also an essential process for alleviating DN. This reactivation of autophagic activity was noted in both in vitro HK-2 cells and diabetic (db/db) mouse models [72,73]. In a recent in vitro study, dapagliflozin restored autophagic activity in high glucose-treated HK-2 cells [73]. It was then suggested that dapagliflozin, as well as SGLT2 knockdown, reversed high glucose-induced autophagic suppression by restoring the suppressed AMPK pathway in HK-2 cells. The activation of AMPK also suppressed NF-κB p65 translocation, and the subsequent decrease in inflammatory cytokine expression suggested a protective effect. Another study by Korbut et al. demonstrated that empagliflozin could ameliorate mesangial proliferation in diabetic mouse kidneys [72]. The authors also observed a normalized renal expression of the apoptosis regulators caspase-3 and Bcl-2, which they proposed as a possible mechanism behind the increase in autophagic activity. The renoprotective effect of SGLT2i reported in these studies persists even without resolution regarding hyperglycemic circumstances, which supports the notion of a glucose-independent effect in kidneys. 

Based on current results, autophagy likely plays a role in the effects of SGLT2i on DKD prevention, and the glucose-independence of SGLT2i further broaden the utility of the agents beyond glucose control. Combined with their cardiovascular benefits, SGLT2i have demonstrated a crucial role in DM and DKD management.
DPP4i and GLP-1 RA also display glucose-independent effects

DPP4i and GLP-1 RA are antihyperglycemic agents that are often used in various combination therapies, and the mechanism of action of both compounds involves the intricate regulatory cascade of the incretin system. The pharmacological effect of DPP4i leads to an increased concentration of GLP-1 and glucose-dependent insulinotropic polypeptide, which are both active incretin hormones, whereas GLP-1 RA directly activates GLP-1 receptors in the pancreas, leading to increased insulin release and reduced glucagon [74,75]. Previous evidence showed that DPP4i displays antifibrotic properties and the ability to inhibit endothelial-to-mesenchymal transition, and that these direct effects assisted in preserving renal function [76]. Among the DPP4i family, linagliptin stands out in the field of nephrology due to its extra-renal clearance property. Korbut et al. demonstrated that the autophagy-activating and renal function-preserving effects of dapagliflozin in db/db mice was also achieved by linagliptin, both as a monotherapy or in combination therapy [72]. However, few studies have explored the effects of DPP4i on autophagy, and the potential underlying mechanism is also poorly understood. It is also worth noting that clinically, the role of DPP4i in DKD progression has been limited due to their intermediate blood glucose reduction efficacy and neutral effects on cardiovascular/weight change, making them less favorable when compared with other agents. However, like SGLT2i, the attenuation of glomerular change in current findings was reported even in the absence of antihyperglycemic activity, implying that it is critical to understand the mechanism behind DPP4i glucose-independent autophagy regulation. Lastly, GLP-1 RA are known for their cardiovascular benefits, while maintaining a high efficacy with regards to blood glucose reduction. GLP-1 RA share similar signaling events with the pathways of metformin and SGLT2i, with Yang et al. reporting increased autophagic activity after liraglutide administration via activation of the AMPK—mTOR axis [77]. This effect led to reduced oxidative stress and preserved kidney function in diabetic mice, which was also independent of glucose control, according to their data. Combined with a lower risk of developing hypoglycemia or other adverse events, GLP-1 RA exhibit favorable traits for blood glucose control.

In conclusion, many of the most common antihyperglycemic agents exhibit signs of being able to regulate autophagic activity in diabetic kidneys. While most share similar signaling axes, their individual clinical utility will depend on the glucose-independent effects they possess.

### 4.2. RAAS Blockade

The role of RAAS blockade in DKD management

RAAS is known to be a key regulatory mechanism in the pathophysiological process of DKD. Components of the RAAS such as angiotensin II (Ang II) were found to contribute to renal cell hypertrophy during the development of DKD, and further induced renal inflammation [78]. Thus, RAAS blockade agents, such as angiotensin-converting enzyme inhibitors (ACEi), angiotensin receptor blockers (ARB), and mineralocorticoid receptor antagonists (MRA), have been mainstream therapies for treating patients with DM and DKD. 

RAAS was found to be linked to autophagy regulation via studies using various disease models. Gao et al. showed that aldosterone may exert an inhibitory effect on AMPK-mediated autophagy in a high phosphate-induced vascular calcification model [79], and that angiotensin IV suppressed excessive FOXO1-induced autophagy in diabetic cardiomyopathy mice [80]. Wang et al. reported that in an aldosterone-induced mesangial cell proliferation and injury kidney model adaptive autophagy was observed via the responsive activation of the FOXO1 signaling pathway [81]. While innate autophagy activation is known to play a role in RAAS-induced renal stress, there have been limited efforts to date to explore if RAAS blockade agents can directly affect autophagy in the kidney. One connection between RAAS blockade and autophagy was made through a phenomenon known as the aldosterone escape effect, where long term usage of RAAS blockade agents can fail to continuously suppress serum aldosterone levels [82]. Dong et al. showed that spironolactone, a first-generation mineralocorticoid receptor antagonist (MRA), treatment can partially block the RAAS and promote autophagy in podocytes, which led to improved renal outcomes in a DN rat model. The authors proposed using spironolactone to counter the aldosterone escape effect [83].

In summary, as RAAS blockers are a first-line medication widely prescribed to diabetic and hypertensive patients, the relationship between RAAS blockade agents and autophagy in the kidney should be further explored. A better understanding of the RAAS—autophagy axis may impact management strategies and future perspectives on how these medications affect DKD.

### 4.3. Novel Therapeutic Agents

To further address DKD management, new means of pharmacological intervention are necessary to tailor treatments to the needs of different patient groups. Several novel therapeutic agents have been reported to exert varying effects on autophagy in DKD progression. 

#### 4.3.1. Klotho

Klotho is a protein known for its complex and varying roles in multiple organ systems. First characterized in the kidney, Klotho rose to prominence as a regulator of the metabolic pathway of chronic kidney disease—mineral and bone disease (CKD-MBD) [84]. In recent years, Klotho was found to exert a multifaceted impact on kidney disease modulation [85,86,87]. Klotho’s renoprotective effects in DKD progression have been reported by multiple studies over the years, and these effects have been primarily associated with the Wnt/β-catenin, FGF23, and the Nrf2 signaling pathways [88,89,90,91]. Considering its involvement in cellular homeostasis, a correlation between Klotho and autophagy promotion was first identified in an ischemia-reperfusion injury mouse model, where the protein restored autophagic flux and mitigated renal fibrosis progression [92]. This effect was believed to be related to a central regulator of autophagy, the Beclin 1/Bcl2 complex. Klotho was also reported to reduce excessive autophagic activity in a tacrolimus-induced injury mouse model [93]. Recently, Xue et al. demonstrated that Klotho-mediated autophagy also played an important role in DKD progression. The authors showed that Klotho overexpression induced AMPK activation and extracellular signal-regulated kinase (ERK) inhibition, improved tubular cell autophagy, and protected against DKD, demonstrating the potential of Klotho as a therapeutic target [94]. It is also known that the Klotho protein can be divided into an endogenous membrane-bound form and/or a secreted soluble form [95]. Therefore, further studies are imperative to help distinguish the respective effects of the various Klotho isoforms on autophagy in DKD.

#### 4.3.2. Vitamin D/Vitamin D Receptor

Studies regarding the vitamin D/vitamin D receptor (VD/VDR) signaling pathway have been conducted in various disease models. In addition to regulating calcium and phosphorus homeostasis, VDR signaling also plays an important role in kidney disease. Early reports showed that vitamin D levels are significantly lower in DM patients with DKD compared to those without DKD, but that this does not result from worsened renal function [96]. It was then proposed that vitamin D may exert an antidiabetic effect by increasing intracellular calcium concentrations, resulting in activation of β-cell calcium-dependent endopeptidase and subsequently enhanced insulin secretion [97]. Studies have provided evidence of a renoprotective effect of VD/VDR signaling, which was believed to be achieved via inhibition of inflammation [98]. Furthermore, recent data also demonstrate a newfound regulatory relationship between VD/VDR and autophagy. This interaction was discovered in both podocytes and tubular epithelial cells, via the autophagy-related protein Atg16L1 and the AMPK pathway, respectively [99,100]. The expression of Atg16L1 protein is suppressed under high glucose conditions and restored by VDR signaling to normalize autophagosome formation. Active vitamin D3 (aVitD3; 1,25-OH vitamin D3) treatment in diabetic rat models was shown to alleviate DKD progression, which provides a promising outlook for developing a vitamin D-based therapy. To build upon the many known benefits of vitamin D supplementation, thorough studies on the renoprotective properties of vitamin D should receive more attention.

#### 4.3.3. Rapamycin

Rapamycin is a potent immunosuppressive drug that is usually used to avoid rejection in patients who undergo organ transplantation. Rapamycin achieves its function via inhibition of mTOR [101]. As mTOR takes part in various physiological activities, its inhibition by rapamycin can have a wide range of effects. An early report in 2009 demonstrated that systemic administration of rapamycin in diabetic mice can alleviate pathological changes and functional decline of the kidney [102]. One recent study confirmed this effect, while identifying the inhibition of the mTOR-S6K1-LC3II signaling pathway in podocytes as the primary mechanism of action [103]. However, despite promising results, the severe side effects of rapamycin may limit its clinical viability as a therapeutic agent. Several studies have thus focused on plant-derived nutritional supplements [104,105,106] or the essential circadian hormone melatonin [107,108]. However, more studies are needed to qualify these compounds as viable treatment options.

In summary, the use of OHAs has already been shown to have multiple beneficial effects in DM and DKD patients, and a better understanding of the potential glucose-independent effects of these compounds may shed new light on directions for their usage. Similarly, more research into the relationship between RAAS blockade agents and autophagy in the kidney will reveal novel perspectives on the usage of these medications for DKD. Finally, current evidence provides support for the feasibility of novel therapeutic targets, such as Klotho, VD/VDR, and rapamycin, but information regarding the mechanism of action of these compounds will be necessary for expanding the spectrum of DKD treatment. 

## 5. Excessive Autophagy and DKD Progression

As mentioned in the previous section, sufficient activation of autophagy in the early stages of CKD is believed to help balance an excessive inflammatory response and prevent further tissue damage. Indeed, this “balancing” effect has been a primary research focus in the field. However, recent publications suggest that the impact of autophagy in DKD appears to extend beyond a protective effect.

The modulation of autophagy in DKD revolves around the pathophysiological components of oxidative stress, intracellular ROS, and AGE metabolism. Due to the previously established role of autophagy in maintaining cellular homeostasis, it was commonly believed that activating autophagic activity can, in turn, alleviate the stress that diabetes places on kidney tissue. However, in a murine model of post-ischemic AKI, Livingston et al. found that after acute injury autophagy was persistently activated during the interstitial fibrosis process in renal tubular cells [109]. Instead of playing a protective role, this activation led to profibrotic phenotypic alterations and subsequent maladaptive repair, which promoted chronic pathological changes. The authors also showed that fibroblast growth factor 2 (FGF2), a well-documented pro-fibrotic cytokine, was selectively induced by the persistent activation of autophagy. The same phenomenon was also identified in post-AKI patients’ renal biopsy samples in the same study. FGF2 is a key component in DN progression, and overexpression of FGF2 in DN patients is believed to lead to epithelial–mesenchymal transition (EMT) and the development of fibrosis [110,111]. This correlation between autophagy activation and FGF2 induction adds to the complexity of autophagy in DN. Additionally, G2/M cell cycle arrest, an indicator of post-injury fibrotic progression, was found to be promoted by autophagy in the same post-ischemic AKI model [109], but was suppressed by Atg-5-mediated autophagy in a unilateral ureteral obstruction (UUO) murine model in a study by Li et al. [112]. This dynamic change in the role of autophagy was discussed in an earlier study using autophagy reporter mice [113], but based on existing evidence it remains difficult to define a precise function of autophagy as a protective or pro-fibrotic regulator. 

The ill-defined functional role of autophagy in DKD also affects current understanding of optimal management strategies for DKD patients. In previous sections, while we discussed how various medications may promote autophagic activity and allay the progression of DKD, several studies have also reported an opposing association between these agents and autophagy. Niu et al. showed that in high glucose-treated human umbilical vein endothelial cells (HUVECs) and a diabetic mouse model metformin was found to downregulate autophagy via the hedgehog signaling pathway [114]. This autophagy suppression effect attenuated endothelial dysfunction after exposure to the high-glucose condition. Another previously discussed class of OHAs, DPP4i, currently lack mechanistic studies regarding their regulation of autophagy in the kidney. However, in an acute pancreatitis-induced lung injury model, it was found that sitagliptin could inhibit excessive autophagy via activating the p62-Keap1-Nrf2 signaling pathway, which achieved an anti-inflammatory effect [115]. As the activation of the Nrf2 pathway has been linked to antioxidation and reduction in kidney fibrosis in recent studies [116], this brings up the question of whether DPP4i can produce similar effects in the kidney. 

As this is an actively evolving area of research, there is very limited literature available to further discuss the potentially conflicting roles of autophagy in DKD development. With further investigation regarding the etiology and stage-specific regulatory functions of autophagy in DKD and kidney fibrosis, we hope to see the future development of precision therapeutic agents for patients with DKD. 

## 6. Conclusions

The healthcare burden of diabetes is, in large part, caused by the development of DKD and subsequent kidney dysfunction. It has become increasingly apparent that mere control of blood glucose or blood pressure is not a sufficient strategy against DKD. As autophagy has become an exciting area of interest, emerging evidence has strengthened the connection between the process and DKD, while yielding promising candidate therapeutic targets. A close regulation of autophagy in DKD patients may be a future prospect in the field of precision medicine, and further exploration of the matter is essential for curtailing and preventing the progression of DKD.

## Figures and Tables

**Figure 1 cells-12-02691-f001:**
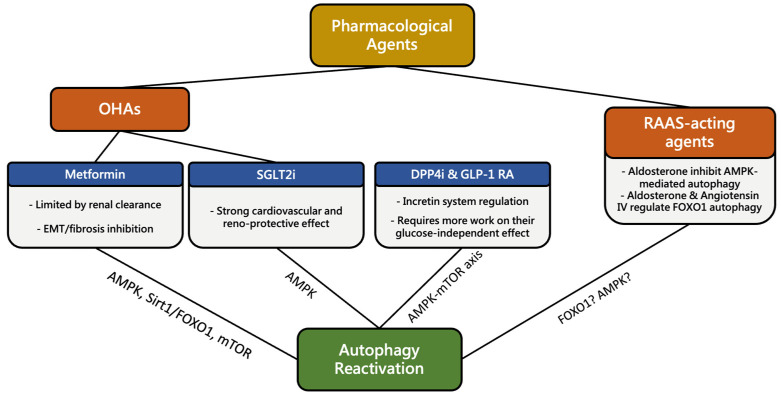
Summary of the correlation between common pharmacological agents and autophagy reactivation. Oral hypoglycemic agents (OHAs) and renin-angiotensin-aldosterone system (RAAS) blockade agents were found to regulate autophagy reactivation in DKD, while reports on the direct impact of RAAS-blockade agents on autophagy in DKD are still lacking. EMT, epithelial—mesenchymal transition; SGLT2i, sodium–glucose cotransporter 2 inhibitors; DPP4i, dipeptidyl peptidase-4 inhibitors; GLP-1 RA, glucagon-like peptide-1 receptor analogs; AMPK, AMP-activated protein kinase; Sirt1, Sirtuin-1; FOXO1, Forkhead box protein O1; mTOR, mechanistic target of rapamycin.

**Table 1 cells-12-02691-t001:** Summary of autophagy-related pathways.

Pathway	Effects on Autophagy	References
JAK/STAT	Activated in DKD mice and suppresses autophagy, reversed by JAK inhibitor in vitro.	[19]
P53	Activated in AGE-treated podocytes, inducing miR-34a transcription, which, in turn, suppresses SIRT1 and enhances p53 activation. Activated in diabetic patients, downregulates ULK-1 and autophagy.	[20,21]
P2Y2R	Downregulates SIRT-1, leading to suppressed autophagy in DN mice.	[22]
Macrophage	Hyperglycemic state suppresses autophagy in macrophages, promotes migration and adhesion, and alters macrophage polarization.	[23,24]
HDACs	Activated in DM patients, mice models, and AGE-treated podocytes, modulates podocyte motility and suppresses autophagy.	[25]

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
