# Peer review of "The Role of Autophagy in Type 2 Diabetic Kidney Disease Management"

_cells, 2023, doi:10.3390/cells12232691_

Round 1
Reviewer 1 Report (Previous Reviewer 2)
Comments and Suggestions for Authors
The authors responded well to my previous comments. I have no further points.
Author Response
We sincerely thank the reviewer for this comment.
Reviewer 2 Report (Previous Reviewer 1)
Comments and Suggestions for Authors
Authors have improved language and added figures to improve the review. No further comments or suggestions.
Author Response
We sincerely thank the reviewer for this comment.
Reviewer 3 Report (New Reviewer)
Comments and Suggestions for Authors
This investigation addresses a relevant issue, considering the high prevalence of diabetes mellitus among the world population. The review addresses the main mechanisms and signaling pathways related to the modulation of autophagy, as well as reporting the impact of hyperglycemia on this process and its relationship with kidney damage, with the aim of elucidating the role of autophagy in the progression of DKD. The review also provides an overview of the effects of the main pharmacological agents on these events.
Title: It would be interesting to include the type of diabetes considered as the focus of the investigation (both in the “Title” and in the “Abstract”).
Abstract: The abstract is well described, but it fails to provide a synthesis, even if brief, of the main findings of studies on the subject, that is, the main impacts of autophagy on DKD, as well as to provide a more specific conclusion. The abstract also does not mention that this review provides an investigative approach relating autophagy and available pharmacological agents.
The approach to the key points of the investigation seems confusing.
Some considerations could complement the manuscript:
- Aspects related to diabetes and its complications (definition, types of diabetes, pathophysiological aspects of the disease, oxidative stress, inflammation and the role of advanced glycation end products-AGEs) could be approached in more detail in the “Introduction” or in specific topics that could be added to the review.
- Similarly, molecular aspects related to autophagy (cascade of events involving the autophagy process, changes in macromolecules and organelles) and the relationship between autophagy and pathophysiological disorders resulting from diabetes could be better discussed in the manuscript.
- Although the manuscript focused on the role of autophagy in the progression of DKD resulting from type 2 diabetes, it would be interesting to clarify whether there could be differences in the modulation or role of autophagy in nephropathy resulting from type 1 diabetes (in relation to type 2 ).
- Topic 5 could be complemented with more studies that investigated the effects of autophagy in DKD. The findings of these studies could be summarized in a Table in order to provide an overview of the data available in the literature.
- Check the formatting of the references according to the Journal's guidelines.
Author Response
- Comment 1: Title: It would be interesting to include the type of diabetes considered as the focus of the investigation (both in the “Title” and in the “Abstract”).
- Response 1: We thank the reviewer for their kind words, and for their suggestion to include the type of diabetes considered in the Title and Abstract. We have revised our wording in the Title and Abstract to indicate our focus on discussing the impact of autophagy on Type 2 diabetes mellitus (T2DM). The changes we made to the relevant portion of the Abstract are bolded below:
- Lines 14-15: “Diabetic kidney disease (DKD), or diabetic nephropathy (DN), is one of the most prevalent complications of type 2 diabetes mellitus (T2DM) and causes severe burden on the general welfare of T2DM patients around the world.”
-
Comment 2: Abstract: The abstract is well described, but it fails to provide a synthesis, even if brief, of the main findings of studies on the subject, that is, the main impacts of autophagy on DKD, as well as to provide a more specific conclusion. The abstract also does not mention that this review provides an investigative approach relating autophagy and available pharmacological agents.
-
Response 2: We thank the reviewer for this comment about the Abstract. We have revised the Abstract so that it now outlines our approach and provides a brief synthesis of our main findings. We also indicate in the Abstract that this review highlights the relationship between autophagy and available pharmacological agents, as per the suggestion. The changes we made to the relevant portion of the Abstract are bolded below:
- Lines 20-25: “Typically, autophagic activity plays a role in DKD progression mainly by promoting an inflammatory response to tissue damage, while both overactivated and downregulated autophagy worsen disease outcomes in different stages of DKD. This correlation demonstrates the potential of autophagy as a novel therapeutic target for the disease, and also highlights new possibilities for utilizing already available DN-related medications.”
-
Comment 3: The approach to the key points of the investigation seems confusing. Some considerations could complement the manuscript:
- Aspects related to diabetes and its complications (definition, types of diabetes, pathophysiological aspects of the disease, oxidative stress, inflammation and the role of advanced glycation end products-AGEs) could be approached in more detail in the “Introduction” or in specific topics that could be added to the review.
- Similarly, molecular aspects related to autophagy (cascade of events involving the autophagy process, changes in macromolecules and organelles) and the relationship between autophagy and pathophysiological disorders resulting from diabetes could be better discussed in the manuscript.
-
Response 3: We thank the reviewer for this detailed comment, and agree that adding details about the molecular and pathophysiological aspects of diabetes and autophagy can help make the review more comprehensive. In response, in lines 40-44, we add detail about the pathophysiological mechanism of DKD, as well as the role of reactive oxygen species and advanced glycation end-products. Then, in lines 80-84, we add detail about the events involved in the autophagy process, such as the formation of phagophores and lipid recruitment. In lines 104-108, we add information about the relationship between autophagy and pathophysiological disorders resulting from diabetes, as requested.
The specific changes are bolded below:
Lines 40-44: “The pathophysiological mechanism of DKD arises from the interaction between various metabolic pathways, including reactive oxygen species (ROS) and advanced glycation end-products (AGEs) [2, 4, 5]. This mechanism triggers abnormal downstream cellular responses, which in turn lead to gross vascular injury and renal cell damage.”
Lines 80-84: “The formation of phagophore membranes in mammalian cells appears to be in dynamic equilibrium with other cytosolic membrane structures that primarily originate from the endoplasmic reticulum. The Atg protein family regulates various processes related to autophagic initiation, such as lipid recruitment and increased sensitivity to growth factor and nutrient availability [9].”
Lines 104-108: “Autophagy is known to be connected to the pathophysiological components of diabetes and subsequent complications. Besides mediating the oxidative stress and ROS generated by diabetes, autophagy also inhibits the accumulation of AGEs, a key element involved in the development of DKD [15].”
-
Comment 4: Although the manuscript focused on the role of autophagy in the progression of DKD resulting from type 2 diabetes, it would be interesting to clarify whether there could be differences in the modulation or role of autophagy in nephropathy resulting from type 1 diabetes (in relation to type 2 ).
-
Response 4: We sincerely thank the reviewer for this comment, and do agree that it is important to differentiate between DKD originating from type 2 versus type 1 diabetes. However, we also believe that from a clinical aspect, type 1 and type 2 diabetes have different management strategies and etiologies, making them very different disease entities. In this study, we aimed to provide a specific focus on DKD resulting from type 2 diabetes and further investigate the correlation between autophagy and T2DM pharmacological agents. As we do not want to detract from this focus, we elected not to include information about the differences in the role of autophagy in DKD from T1DM.
-
Comment 5: Topic 5 could be complemented with more studies that investigated the effects of autophagy in DKD. The findings of these studies could be summarized in a Table in order to provide an overview of the data available in the literature.
-
Response 5: We are grateful to the reviewer for this comment.
The purpose of topic 5 (“Excessive autophagy and DKD progression”) in our previous revision is to highlight the new and controversial topic of the impact of excessive autophagy on DKD. To our knowledge, this matter has not been properly addressed in previous review articles, and is not thoroughly studied in the literature. After extensive literature review, we identified a limited number of studies that explore the effects of excessive autophagy, with varying results. The findings of these studies conflict with the existing understanding that autophagy plays a protective role in DKD progression. Instead, the studies that we highlight in topic 5 suggest that autophagy may play a hidden, ill-defined functional role in the later stages of disease development that requires more attention in the future. We believe that compiling a standalone table of this data may not significantly aid readers in better understanding the topic, and instead may lead to a false belief that there is an established scientific consensus with regards to excessive autophagy and DKD. We aimed for topic 5 to be a section that can act as an “epilogue” to our article to trigger further discussion and research on the potentially dynamic role of autophagy along different stages of DKD.
-
Comment 6: Check the formatting of the references according to the Journal's guidelines.
-
Response 6: We thank the reviewer for this comment. We have rechecked our references, and made sure that our formatting fits the Journal’s guidelines as outlined at the following link: https://www.mdpi.com/journal/cells/instructions#references.
Round 2
Reviewer 3 Report (New Reviewer)
Comments and Suggestions for Authors
Adjustments improve the manuscript. Topic 5 could be complemented by the Authors, also highlighting in the study the conflicting or divergent points on the topic, in order to provide an overview of the role of autophagy in DKD, better discussing the modulatory mechanisms and their impact on diabetes, considering the data available in the literature to date.
Author Response
Comment 1: Adjustments improve the manuscript. Topic 5 could be complemented by the Authors, also highlighting in the study the conflicting or divergent points on the topic, in order to provide an overview of the role of autophagy in DKD, better discussing the modulatory mechanisms and their impact on diabetes, considering the data available in the literature to date.
Response 1: We thank the reviewer for this comment. We agree that we can further highlight and summarize the mechanism of autophagy in DKD, and that our topic 5 can provide new insights into the conflicting role of autophagy in DKD. We have added some detail to concisely emphasize the conflicting understanding on the role of autophagy in DKD. However, as there is limited literature available to us to discuss this topic from a quantitative perspective, we took a more speculative approach in our description, as this is a very recent and evolving area of research. The specific changes we made to topic 5 are bolded below.
Lines 468-472: “The modulation of autophagy in DKD revolves around the pathophysiological components of oxidative stress, intracellular ROS, and AGE metabolism. Due to the previously established role of autophagy in maintaining cellular homeostasis, it was commonly believed that activating autophagic activity can, in turn, alleviate the stress that diabetes places on kidney tissue.”
Lines 505-506: “As this is an actively evolving area of research, there is very limited literature available to further discuss the potentially conflicting roles of autophagy in DKD development.”
This manuscript is a resubmission of an earlier submission. The following is a list of the peer review reports and author responses from that submission.
Round 1
Reviewer 1 Report
Comments and Suggestions for Authors
This review by Tseng et al. aims to capture new insights of the role of autophagy in Diabetic Kindey Disease management. The subject is interesting and relevant, although it should be noted there was a very recent review in frontiers in endocrinology covering almost the same subject (DOI 10.3389/fendo.2023.1139444).
Despite the interesting subject, the review was hard to read and requires substantial work to be useful for a more general audience. First, the text would benefit from a major uphaul. Past and present tense is not used correctly in many places. There is superfluous text, long sentences, and passive voice that deter from effectively communicating a clear message. Furthermore, manuscript describes many results and recent work, but does a poor job in integration and interpretation of the data and providing the bigger picture that connects all different aspects. Some mechanisms are superficially described, and several important items lack an introduction, background, or context.
Unfortunately, as a result of all the above, this review did not provide me the comprehensive overview and understanding of the role of autophagy in DKD/ DKD treatment that I would look for in a review and , I am sure, the authors would like to provide (considering their reputation and efforts).
Below are some examples of my concern, which I hope will help the authors.
Also, it seems like there is a missed opportunity by not using more intricate figures to support the review.
Line 17 connections between… AND (not to)?
Line 42 mechanisms? Better a pathway as no mechanism is described.
Line 47 “resulted injury”? Kidney damage?
Line 49 “On the other hand” ? Delete?
Line 51 “could lead” can cause?
Line 51 “Generally, autophagy 51 could be divided into” There are…
Line 53: “ , with macroautophagy” . Macroautophagy is the most prevalent form and will be referred to ..
Line 56 phagosphore => phagophore
Line 59 “decomposition”?
Line 59 “Could interfer” interfere or can interfer . What is the purpose of line 59-60? State that the process is regulated? Or that this process can be messed up? As is seemed unclear in its meaning…
Line 62 past tense.. should be present => indicates & has
Line 63 “could be a rising therapeutic targets in the future” => has potential as a therapeutic target.
Line 68 “to” => that?
“Previous studies have discovered that autophagy pathway regulation is critical in various diseases since it has control over the inflammation 70 and fibrosis response during cellular stress or injury.”
“misregulation of autophagy pathways contributes to various diseases because the autophagy process controls the inflammation 70 and fibrosis response during cellular stress or injury”?
“In the case of kidney diseases, with the aid of gene transcriptional pathway datasets bioinformatics analysis, reactive autophagy was found to be activated in multiple types of glomerular injury, including membranous nephropathy (MN), focal segmental glomerulosclerosis (FSGS), IgA nephropathy 74 (IgAN), and especially in DN [8].” => Awkward and long sentence and what is “reactive autophagy” here?
Line 80 AKI to CKD acute kidney injury? o chronic kidney disease?
Line 89 “being under excessive energy or nutrient” what does that mean?
Line 89-92 rephrase sentence to make it more clear.
Line 93 “which hinders the formation of normal autophagosome.” Hinders or prevent induction of autophagy genes?
Line 94 what do hyperglycemic conditions do to the nutrient state of the cell and the activity of mTOR and AMPK?
Several places the authors use dysregulation like (dysregulation of these pathways in line 94. Are these pathways dysregulated or do they respond as they are designed to respond, based on inadverted signals provided by the hyperglycemic conditions… also if possible being more specific in it being activated or inhibited when dysregulation can be more informative….
Line 96 indicated => indicates. Also what is this evidence?
Line 101 Recent studies helped elucidate the modulation from upstream pathways. Modulation of what?
Line 107 “in the cell models” which cell models, earlier text described a mice model
Line 108 “Combined with previous results regarding the mTOR signaling pathways,” No reference and no discussion of these data?
Line 109 ”the recently introduced cornerstone of autophagic signaling,” what does this mean?
Line 121 “ P2Y2R was found to prominently expressed in the renal glomerular” to be?
Paragraph 3.1 summarizes a few papers showing links between nutrient signaling pathways – autophagy and DN, but no integration or bigger picture is provided.
Line 143 “Sharing similar importance on a larger scale, another important physiological repairing mechanism that has been picking up momentum in the researching field, is the effect of macrophage.” Please rephrase to a more informative and scientific sentence.
Line 159 “the crosstalk between macrophage and autophagy” This does not make sense… cross talk between a cell and a process within that cell? Or the kidney cells?
Line 160 “The authors also reviewed the mesenchymal stem cells (MSCs) therapy in DN mice, with data showing that it promoted macrophage autophagy via activation of transcription factor EB (TFEB),which could explain the underlying mechanism behind the beneficial effect of MSC therapy in DKD [25].” Seems like a lot of steps are assumed or missing and would need further explanation to provide a convincing rational and mechanism..
Section3.2 ends with a concluding sentence that states “macrophage polarization”. However, the section only briefly mentions it regarding M1 cells without further emphasizes or explanation.. makes for a disconnect.
Full of superfluous and non-relevant text line 169 : “With precision medicine being the 169 future trend of healthcare, the recognition of disease-specific genetic component has be- 170 come increasingly crucial. Building onto that, the epigenetic mechanisms, including but 171 not limited to histone modifications, microRNAs regulation, were believed to assist in the 172 physiological adjustments to the given environmental conditions. They were recently 173 brought into the spotlight due to its extensive effect on DKD progression, with the com- 174 ponents of epigenetic modulations, inclusive of noncoding RNAs (ncRNAs), histone 175 deacetylases (HDACs), and DNA methylation, all had their respective role on the disease 176 [26, 27]. “
Epigenetic modulations, including noncoding RNAs (ncRNAs), histone 175 deacetylases (HDACs), and DNA methylation, all have been shown to impact DKD progression (refs).
Line 180 “were identified to have involvement”…. are involved
Line 193 “could achieve…..could be” Did it? Or … we predict… .. suggesting they might be….?
Line 185 HDACs has…. took => HDACs have…. take
Line 199 “such as valproic acid, a widely used anti-epileptic drug, as an example “ redundant word use
Figure 1 should have some legend to make the basics understanadable without main text.
Line 248 “suggested that with lansoprazole “. , like lansoprazole. This results from PPIs blocking autophagy….
Line 252 “Whether the effect could be translated to other medication that would assist in recovering autophagy needs to be confirmed. “ Whether other medications….
Line 260 Why “On the other hand”?
Section 4.1.1. discussed mechanisms regarding SGLT2 and DPP4 inhibitors eby stating some pathways impacted, however SGLT2 and DPP4 functions are not even discussed.. so the discussion is largely descriptive and does not involve mechanism.
Line 264 by both. Reference missing?
Line 310 “the current regimen is still longing for a makeover to breathe new life into 310 the impending doom of DKD progression. “ seems rather poetic for a scientific review.
Line 320 “a multifaceted impact on kidney disease modulation. “ what does this mean?
Line 365 seems prudent to emphasize here it stimulates autophagy
Line 370 “chose to highlight” described or reported
“More studies need to be conducted in order for them to be established as reliable options that are commercially available. “ However, more research is required to establish their potential as suitable drugs.
Comments on the Quality of English Language
see above
Reviewer 2 Report
Comments and Suggestions for Authors
This is a very well and diligently written, up-to-date review about autophagy in diabetic nephropathy. I only have a few minor comments:
line 134-135: please change sentence to a less empathic version ("it is fascinating that there is still much to dig ..." is too much passion for a scientific review).
line 221 to 227ff and fig.1: please expand figure 1 and include more data here. I disagree with the term "OHA" being linked to SGLT2-inhibitors as SGLT2i function different my lowering the influx to the glomerulus via the macula densa and not by lowering blood glucose. SGLT2i also action via sodium excretion. Please revise.
Comments on the Quality of English Language
None.
Reviewer 3 Report
Comments and Suggestions for Authors
Dear Editor,
The authors in this paper have tried to synthesize the new evidence regarding the Role of autophagy in diabetic kidney disease management. The work has a coherent structure and is well written, however, there are similar reviews on the same topic that have been published in these years (e.g. Front. Endocrinol. 2023 https://doi.org/10.3389/fendo.2023.1139444; Cells 2021 https://doi.org/10.3390/cells10092497; Adv Exp Med Biol 2020 https://doi.org/10.1007/978-981-15-4272-5_36; Cell Mol Life Sci 2018 https://doi.org/10.1007/s00018-017-2639-1). Moreover, these reviews have much more solid structure with a greater amount of information and summary images than the review submitted by Che-Hao Tseng et al.
In light of this, the work submitted by the authors is too concise and does not lead to significant updates, therefore I do not recommend publication.
Comments on the Quality of English LanguageModerate editing of English language required